# Esculetin as a Bifunctional Antioxidant Prevents and Counteracts the Oxidative Stress and Neuronal Death Induced by Amyloid Protein in SH-SY5Y Cells

**DOI:** 10.3390/antiox9060551

**Published:** 2020-06-25

**Authors:** Letizia Pruccoli, Fabiana Morroni, Giulia Sita, Patrizia Hrelia, Andrea Tarozzi

**Affiliations:** 1Department for Life Quality Studies, Alma Mater Studiorum—University of Bologna, Corso d’Augusto 237, 47921 Rimini, Italy; letizia.pruccoli2@unibo.it; 2Department of Pharmacy and Biotechnology, Alma Mater Studiorum—University of Bologna, Via Irnerio 48, 40126 Bologna, Italy; fabiana.morroni@unibo.it (F.M.); giulia.sita2@unibo.it (G.S.); patrizia.hrelia@unibo.it (P.H.)

**Keywords:** amyloid beta, oxidative stress, neuronal death, esculetin, bifunctional antioxidants

## Abstract

Oxidative stress (OS) appears to be an important determinant during the different stages of progression of Alzheimer’s Disease (AD). In particular, impaired antioxidant defense mechanisms, such as the decrease of glutathione (GSH) and nuclear factor erythroid 2 (NF-E2)-related factor 2 (Nrf2), a master regulator of antioxidant genes, including those for GSH, are associated with OS in the human AD brain. Among the neuropathological hallmarks of AD, the soluble oligomers of amyloid beta (Aβ) peptides seem to promote neuronal death through mitochondrial dysfunction and OS. In this regard, bifunctional antioxidants can exert a dual neuroprotective role by scavenging reactive oxygen species (ROS) directly and concomitant induction of antioxidant genes. In this study, among natural coumarins (esculetin, scopoletin, fraxetin and daphnetin), we demonstrated the ability of esculetin (ESC) to prevent and counteract ROS formation in neuronal SH-SY5Y cells, suggesting its profile as a bifunctional antioxidant. In particular, ESC increased the resistance of the SH-SY5Y cells against OS through the activation of Nrf2 and increase of GSH. In similar experimental conditions, ESC could also protect the SH-SY5Y cells from the OS and neuronal death evoked by oligomers of Aβ_1–42_ peptides. Further, the use of the inhibitors PD98059 and LY294002 also showed that Erk1/2 and Akt signaling pathways were involved in the neuroprotection mediated by ESC. These results encourage further research in AD models to explore the efficacy and safety profile of ESC as a novel neuroprotective agent.

## 1. Introduction

Several studies suggest that oxidative stress (OS) is an early event in the pathogenesis of Alzheimer’s disease (AD) [1,2]. OS also appears to be an important determinant during the different stages of progression of AD [3]. In this context, it is likely that a causal relationship occurs between OS and several neuropathological hallmarks of AD, including amyloid plaques containing amyloid beta (Aβ) peptides and neurofibrillary tangles at an extra- and intra-cellular level, respectively [4]. Among the different forms of Aß peptides, the soluble Aβ _1–42_ peptides or shorter fragments seem to promote OS through their interaction with redox-active metals [5]. Aβ may further contribute to OS by triggering *N*-methyl-d-aspartate receptor-dependent Ca^2+^ influxes leading to mitochondrial dysfunction with generation of reactive oxygen species (ROS) and subsequent neuronal death [4].

High OS in AD and its prodromal stage, mild cognitive impairment, have been presumed to also be a consequence of compromised antioxidant defense mechanisms in both the blood and brain [6]. Several studies record that the decrease of glutathione (GSH) and GSH synthetase involved in GSH synthesis in the hippocampus and frontal cortex are associated with development of AD pathology [7,8]. These alterations of redox homeostasis are consistent with decreased levels of nuclear factor erythroid 2 (NF-E2)-related factor 2 (Nrf2), a master regulator of antioxidant defense and detoxification genes activated by OS conditions, in hippocampal neurons of human AD brain [9]. More recent studies suggest that the maintenance of redox homeostasis by Nrf2 is able to prevent not only OS but also inflammation, a critical process in the pathogenesis of neurodegenerative diseases, in in vitro and in vivo neuroinflammation models [10]. In particular, Nrf2 represses the inflammatory responses mediated by nitric oxide synthase, interleukin-6, tumor necrosis factor-α. Nrf2 responds differently to ROS generated during acute and chronic OS. In particular, during chronic OS frequently detected in aging and in AD, the Nrf2 pathway becomes unresponsive to ROS [10,11]. Hence, the interplay between Nrf2 activation and chronic OS suggests neuroprotective strategies using an association of compounds that can either activate the Nrf2 pathway by ROS independent mechanisms or act directly as ROS scavenger compounds that may thus be required to reduce OS in AD [12]. In this regard, several bifunctional antioxidants, such as lipoic acid, curcumoids, chalcones, polyenes, and phenolic Michael acceptors can exert the ability both to scavenge oxidants directly (i.e., direct antioxidant activity) and to induce the expression of cytoprotective phase 2 genes which have different antioxidant actions (i.e., indirect antioxidant activity) [13].

Coumarins are a class of phenolic compounds found in several medicinal plants, for example *Cichorium intybus*, *Artemesia capillaris*, *Ceratostigma willmottianum* and *Citrus limonia* [14,15]. Among the coumarins, esculetin, ESC (6,7-dihydroxycoumarin) has been shown to have interesting pharmacological actions in dysmetabolic syndromes, cardiovascular diseases, renal dysfunctions, cancer and neurological disorders [16]. In this context, recent studies showed the ability of ESC to prevent and counteract OS, mitochondrial dysfunction, inflammation and neuronal death, in different animal models of psychiatric disorders, cerebral ischemia and Parkinson’s disease [17,18,19,20,21]. Taken together, these studies highlight the ability of ESC to cross the blood–brain barrier (BBB) and to exert neuroprotective effects at the brain level against several pathological conditions. However, the potential antioxidant and neuroprotective effects of ESC against neurodegenerative events associated with AD still remain unanswered.

In this study, we evaluated the ability of ESC to prevent and counteract OS as well as the neuronal death elicited by oligomers of Aβ_1–42_ peptides (OAβ_1–42_), soluble aggregates of Aβ peptides involved in the pathogenesis of AD, in neuronal SH-SY5Y cells. In particular, we initially assessed the antioxidant effects of ESC together with other natural coumarins, including scopoletin (SCOP), fraxetin (FRAX) and daphnetin (DAPH), that show a similar chemical structure (Figure 1), in order to also evaluate the relationship between the chemical structure of these coumarins and the direct and indirect antioxidant properties at neuronal levels. The experimental approach was thus characterized by different treatments of SH-SY5Y cells with ESC, before or during the neuronal damage evoked by OAβ_1–42_, to appraise the ability of ESC to prevent or counteract pathogenic events of AD.

## 2. Materials and Methods

### 2.1. Chemicals

The coumarins ESC (purity: ≥98%), SCOP (≥99%), FRAX (≥98%) and DAPH (≥97%), 2,2’-Azino-bis (3-ethylbenzothiazoline-6-sulfonic acid) diammonium salt (ABTS), 2’,7’-dichlorodihydrofluorescein diacetate (DCFH-DA), *tert*-butyl hydroperoxide solution (*t*-BuOOH), monochlorobimane (MCB), dihydroethidium (DHE), 3-(4,5-dimethyl-2-thiazolyl)-2,5-diphenyl-2H-tetrazolium bromide (MTT), 2,2-diphenyl-1-picrylhydrazyl (DPPH), propidium iodide (PI), PD98059 (PD), DL-Buthionine-(S,R)-sulfoximine (BSO) and anti-β-actin antibody were purchased from Sigma-Aldrich (Sigma-Aldrich, St. Louis, MO, USA). LY294002 (LY) was purchased from Alexis Biochemicals (Alexis Biochemicals, San Diego, CA, USA). The Nrf2 antibody was purchased from Santa Cruz (Santa Cruz Biotecnology, Dallas, TX, USA). Akt (serine–threonine kinase), Phospho-Akt, Erk1/2 (extracellular signal-regulated kinase), Phospho-Erk1/2, GSK3β (glycogen synthase kinase-3β), Phospho-GSK3β and Lamin B1 antibodies were purchased from Cell Signaling (Cell Signaling Technology, Danvers, MA, USA). Beta-Amyloid (1-42) peptide was purchased from AnaSpec (AnaSpec, Fremont, CA, USA). The Nuclear Extract and TransAM Nrf2 Kit were purchased from Active Motif (Active Motif, Carlsbad, CA, USA). All chemicals used were of high purity analytical grade.

### 2.2. Cell Culture and Preparation of Coumarin Solutions

Human neuronal SH-SY5Y cells were purchased from the Lombardy and Emilia Romagna Experimental Zootechnic Institute (Italy). SH-SY5Y cells were routinely grown in Dulbecco’s modified Eagle Medium with phenol red supplemented with 10% fetal bovine serum, 2 mM L-glutamine, 50 U/mL penicillin and 50 μg/mL streptomycin at 37 °C in a humidified incubator with 5% CO_2_. For the experiments with SH-SY5Y cells, stock coumarin solutions were prepared in dimethyl sulfoxide (DMSO) at 20 mM. The stock solutions were further diluted in complete medium to obtain the desired concentrations of coumarins in a maximum of 0.1% DMSO.

### 2.3. Determination of Neuronal Viability

The neuronal viability was evaluated by the reduction of MTT to its insoluble formazan, as previously described [22]. Briefly, SH-SY5Y cells were seeded in a 96 well plate at 2 × 10^4^ cells/well, incubated for 24 h and subsequently treated with various concentrations of the studied coumarins (2.5–80 µM) for 24 h at 37 °C in 5% CO_2_. The treatment medium was then replaced with MTT in Hank’s Balanced Salt Solution (HBSS) (0.5 mg/mL) for 2 h at 37 °C in 5% CO_2_. After washing with HBSS, formazan crystals were solubilized in isopropanol. The amount of formazan was determined (570 nm, reference filter 690 nm) using the multilabel plate reader VICTOR™ X3 (PerkinElmer, Waltham, MA, USA). Data are expressed as percentage relative to untreated cells.

### 2.4. Determination of Intrinsic Antioxidant Activity

The intrinsic antioxidant activity of coumarins was determined using DPPH and ABTS radicals. The DPPH assay was performed as previously described in [23]. Briefly, 150 µL of DPPH in ethanol (100 µM) was added to 50 µL of coumarins at different concentrations (2.5–40 µM) in a 96 well plate. After 30 min of incubation, the absorbance of the reaction solution was measured at 490 nm using the multilabel plate reader VICTOR™ X3 (PerkinElmer, Waltham, MA, USA).

The ABTS assay was performed as previously described in [24]. Briefly, ABTS radical was generated by mixing a 2 mM ABTS solution with 7 mM potassium persulfate (K_2_S_2_O_8_) and incubating in the dark for 24 h at room temperature. Before use, the ABTS solution was diluted (1–25 mL of phosphate buffered saline) to obtain an absorbance value of 0.70 ± 0.02 at 734 nm. Upon addition of 1 mL of the diluted ABTS solution to 10 µL of the studied coumarins at different concentrations (2.5–40 µM), the absorbance at 734 nm was recorded after 1 min. The antioxidant activity was expressed as a concentration of coumarin able to decolorize 50% of the ABTS radical.

### 2.5. Determination of Direct and Indirect Antioxidant Activity

The direct and indirect antioxidant activity of coumarins was assessed in SH-SY5Y cells as previously described in [25]. Briefly, SH-SY5Y cells were seeded in a 96 well plate at 2 × 10^4^ cells/well, incubated for 24 h and subsequently treated with coumarins at different concentrations (2.5–20 µM) for 2, 6, 12 and 24 h at 37 °C in 5% CO_2_. The treatment medium was then discarded and 100 µL of the fluorescent probe DCFH-DA in phosphate buffered saline (10 g/mL) was added to each well. After 30 min of incubation at room temperature, the DCFH-DA solution was replaced with 100 µL of *t*-BuOOH (200 µM). After further 30 min, the ROS formation was measured (excitation at 485 nm and emission at 535 nm) using the multilabel plate reader VICTOR™ X3 (PerkinElmer, Waltham, MA, USA). Data are expressed as fold increases in ROS formation evoked by *t*-BuOOH.

### 2.6. Determination of Antioxidant Coumarins in Membrane and Cytosolic Fractions

The cellular uptake was determined indirectly by using the ABTS assay, as previously described in [26]. Briefly, SH-SY5Y cells were seeded in 60 mm dishes at 2 × 10^6^ cells/dish, incubated for 24 h and subsequently treated with ESC and DAPH (20 µM) for 2 h at 37 °C in 5% CO_2_. At the end of incubation, cells were washed 3 times with cold phosphate buffered saline (PBS) and removed from the dish by gently scraping with a cell lifter. Cells were then collected in 1 mL of PBS and centrifuged at 10,000 rpm for 10 min at 4 °C. The supernatant was removed, and cells were washed with 1 mL of PBS. This was repeated a further 2 times and the pellet was finally reconstituted in 600 µL of lysis buffer containing Triton X-100 0.05%. Cells were then homogenized and allowed to stand at 4 °C for 30 min. The cytosolic fraction was obtained by centrifugation at 14,000 rpm for 15 min at 4 °C. The remaining pellet was solubilized in 400 µL of lysis buffer containing Triton X-100 1% to obtain the membrane fraction. Cytosolic and membrane fractions were stored at −20 °C. Small amounts were removed for the determination of the protein concentration according to the Bradford method. The antioxidant activity of coumarins was then measured on cytosolic and membrane fractions using the ABTS assay. The final total antioxidant activity (TAA) of cytosolic and membrane fractions was calculated by comparing ABTS decolorization with that of a standard curve of Trolox, a water-soluble analog of vitamin E. The standard curve was prepared using Trolox in a range of concentrations from 50 to 150 µM. Data are expressed as micromole of Trolox equivalent antioxidant activity per milligram of protein (µmolTE/mg protein).

### 2.7. Determination of GSH Levels

GSH levels were assessed in SH-SY5Y as previously described [27]. SH-SY5Y cells were seeded in a black 96 well plate at 2 × 10^4^ cells/well, incubated for 24 h and subsequently treated with ESC and DAPH (20 µM) for 1–24 h at 37 °C in 5% CO_2_. At the end of incubation, the treatment medium was discarded and 100 µL of the fluorescent probe MCB in PBS (50 µM) were added to each well. After 30 min of incubation at 37 °C, GSH levels were measured (excitation at 355 nm and emission at 460 nm) using the multilabel plate reader VICTOR™ X3 (PerkinElmer, Waltham, MA, USA). Data are expressed as fold increases versus untreated cells.

### 2.8. Nuclear Extraction and Determination of Nrf2 Nuclear Levels

Nrf2 nuclear levels were evaluated in SH-SY5Y cells as previously described in [28]. SH-SY5Y cells were seeded in 60 mm dishes at 2 × 10^6^ cells/dish, incubated for 24 h and subsequently treated with ESC and DAPH (20 µM) for 1 and 3 h at 37 °C in 5% CO_2_. At the end of incubation, nuclear extraction and determination of Nrf2 nuclear levels were performed using Nuclear Extract Kit (Active Motif) and Western blotting, respectively.

The nuclear extracts (50 µg per sample) were separated by 4–15% SDS polyacrylamide gels and transferred onto nitrocellulose membranes, which were probed with primary Nrf2 antibody (1:1000) and secondary antibody. Enhanced chemiluminescence reagents (Pierce, Rockford, IL, USA) were utilized to detect targeted bands. The same membranes were stripped and re-probed with β-actin and Lamin B1 antibodies Data were analyzed by densitometry, using Quantity One software (Bio-Rad Laboratories S.r.L., Hercules, CA, USA). Data are expressed as fold increases versus untreated cells.

### 2.9. Determination of Erk, Akt and GSK3β Protein Phosphorylation

The phosphorylation of Erk, Akt and GSK3β kinases was evaluated by using the Western blotting method. SH-SY5Y cells were seeded in 60 mm dishes at 2 × 10^6^ cells/dish, incubated for 24 h and subsequently treated with ESC (20 µM) for 15, 30, 60 and 120 min at 37 °C in 5% CO_2_. At the end of incubation, cells were trypsinized and the cellular pellet was resuspended in complete lysis buffer containing leupeptin (2 µg/mL), PMSF (100 µg/mL) and cocktail of protease/phosphatase inhibitors (100×). Small amounts were removed for the determination of the protein concentration using the Bradford method. The protein lysates (30 μg per sample) were separated by 4–15% SDS polyacrylamide gels (Bio-Rad Laboratories S.r.L.) and transferred onto 0.45 μm nitrocellulose membranes, which were probed with primary phosphorylated antibody p-Erk, p-Akt and p-GSK3β (all 1:1000), and secondary antibodies. ECL reagents (Pierce, Rockford, IL, USA) were utilized to detect targeted bands. The same membranes were stripped using buffer containing β-mercaptoethanol (18.3 μM), SDS (69.35 mM), and Tris·HCl (62.5 mM), pH 6.7, and then re-probed with Erk, Akt and GSK3β antibodies (all 1:1000). Data were analyzed by densitometry, using Quantity One software (Bio-Rad Laboratories S.r.L.). Data are expressed as the ratio between phosphorylated form and total protein expression.

### 2.10. Aβ_1–42_ oligomer Preparation

Aβ_1–42_ peptide was first dissolved in hexafluoroisopropanol to 1 mg/mL, sonicated, incubated at room temperature for 24 h and lyophilized. The resulting unaggregated Aβ_1–42_ peptide film was dissolved with dimethyl sulfoxide and stored at −20 °C until use. The Aβ_1–42_ peptide aggregation to oligomeric form was prepared as previously described [29].

### 2.11. MTT Formazan Exocytosis Assay

The neuroprotective activity of ESC against Aβ_1–42_ oligomers was evaluated in SH-SY5Y cells using the MTT formazan exocytosis assay [30]. Briefly, SH-SY5Y cells were seeded in a 96 well plate at 3 × 10^4^ cells/well, incubated for 24 h and subsequently incubated with ESC (20 µM) and OAβ_1–42_ (10 µM) for 4 h at 37 °C in 5% CO_2_. At the end of incubation, the treatment medium was replaced with MTT in HBSS (0.5 mg/mL) for 1 h at 37 °C in 5% CO_2_. Intracellular MTT granules were first solubilized by 1% Tween 20 at 37 °C for 10 min with shaking. Solubilized formazan in the supernatant was then transferred to a new plate as the Tween 20-soluble MTT (TS-MTT). The remaining cell surface needle-like crystals were solubilized with 100% Isopropanol as the Tween 20-insoluble MTT (TI-MTT). Absorbance values at 590 nm were determined for each fraction using 630 nm as the reference wavelength. Data are expressed as percentages with control set at 100%.

### 2.12. Determination of ROS Formation Induced by Aβ_1–42_ Oligomers

To evaluate the antioxidant activity of ESC against ROS formation induced by OAβ_1–42_, SH-SY5Y cells were seeded in a 96 well plate at 5 × 10^3^ cells/well, incubated for 24 h and subsequently treated with ESC (20 µM) for 24 h and Aβ_1–42_ oligomers for 3 h at 37 °C in 5% CO_2_. At the end of incubation, the treatment was removed and 100 µL of the fluorescent probe DHE (10 µM) were added to each well. After 30 min of incubation at 37 °C in 5% CO_2_, the probe DHE was replaced with HBSS and ROS formation was determined using the fluorescence microscope Eclipse Ti-E (Nikon Instruments Spa, Florence, Italy) equipped with TRITC filters (EX 535/50, BS 575, EM 590LP), DS-U3 camera and NIS-Elements BR 3.2 64-bit Software. The intensity of fluorescence was measured from an area corresponding to 20 cells in at least five different random areas. Data are expressed as arbitrary units of fluorescence (AUF).

### 2.13. Determination of Neuronal Death Induced by Aβ_1–42_ Oligomers

To evaluate the ability of ESC to counteract the neuronal death induced by OAβ_1–42_, SH-SY5Y cells were seeded in a 96 well plate at 5 × 10^3^ cells/well, incubated for 24 h and treated with ESC (20 µM) and Aβ_1–42_ oligomers (10 µM) for 24 h at 37 °C in 5% of CO_2_. To evaluate the ability of ESC to prevent the neuronal death induced by OAβ_1–42_, cells were treated with ESC (20 µM) for 24 h and subsequently with OAβ_1–42_ (10 µM) for 24 h. In parallel, cells were treated with ESC [20 µM] for 24 h in the absence or presence of BSO (400 µM), PD98059 (5 µM) and LY294002 (10 µM) and then with OAβ_1–42_ (10 µM) for 24 h. At the end of incubation, the neuronal death was determined using the fluorescent probe PI (25 µg/mL) and the fluorescence microscope Eclipse Ti-E (Nikon Instruments Spa). The total cells were counted in a bright field then only the red dead cells were counted using TRITC filters (EX 535/50, BS 575, EM 590LP). Data are expressed as percentages of dead cells versus total cells.

### 2.14. Statistical Analysis

Data are reported as mean ± standard deviation (SD) of at least three independent experiments. Statistical analysis was performed using one-way ANOVA with the Dunnett or Bonferroni post-hoc test. Differences were considered significant at *p* < 0.05. Analysis was performed using PRISM 5 software (GraphPad Software, La Jolla, CA, USA).

## 3. Results and Discussion

### 3.1. Direct and Indirect Antioxidant Activity of Coumarins

The antioxidant activity of the studied coumarins, in terms of ability to scavenge the free radical, was initially evaluated using ABTS assays and expressed as IC_50_ (concentration of coumarin able to neutralize 50% of the ABTS radical). The coumarins were shown to neutralize the ABTS radicals according to the following order of potency: ESC (IC_50_: 2.53 ± 0.26 µM) > DAPH (IC_50_: 13.01 ± 0.10 µM) > FRAX (IC_50_: 14.35 ± 0.06 µM) > SCOP (IC_50_: 18.75 ± 0.26 µM). ESC, DAPH and FRAX also exerted a similar antioxidant activity against DPPH radical (Appendix A).

In this regard, several studies report that the orto-dihydroxyl (catechol) group and the α-pyrone ring present in coumarins ESC, FRAX and DAPH contribute to their radical scavenger and antioxidant activity [31,32]. Other studies also suggest that the number of hydroxyl groups on the ring structure of coumarins is strictly correlated with their antioxidant effects [33]. In agreement with this evidence, SCOP recorded the lowest scavenger activity compared to that of the other studied coumarins.

To evaluate the antioxidant activity of ESC, FRAX, DAPH and SCOP in SH-SY5Y cells, we firstly established the range of coumarin concentrations not associated with neurotoxicity. The treatment of the SH-SY5Y cells with concentrations up to 20 µM did not affect neuronal viability using the MTT assay (data not shown). The range of concentrations 5–20 µM was thus selected for all the following experiments. To discriminate the direct and indirect antioxidant activity of the studied coumarins, the SH-SY5Y cells were treated for 2 h and 24 h with the respective coumarins, and then treated with *t*-BuOOH, a lipophilic hydroperoxide that shows the ability to generate ROS from lipid peroxidation in brain tissues [34]. At the end of the treatments, the ROS formation was determined using the fluorescent probe DCFH-DA. The treatment of SH-SY5Y cells for 2 h with 10 and 20 µM of ESC, FRAX and DAPH significantly counteracted the ROS formation induced by *t*-BuOOH [200 µM] for 30 min, while SCOP did not show any antioxidant activity (Figure 2). The ability of ESC, FRAX and DAPH, but not SCOP, to counteract the ROS formation was also significantly correlated to their ability to scavenge the ABTS radical at different concentration levels (ESC: *R*^2^ = 0.83, *p* < 0.05; FRAX: *R*^2^ = 0.93, *p* < 0.01; DAPH: *R*^2^ = 0.78, *p* < 0.05) confirming the contribution of the catechol group to antioxidant effects recorded in SH-SY5Y cells. In contrast to FRAX and SCOP, ESC and DAPH at 20 µM also showed the ability to significantly prevent the ROS formation induced by *t*-BuOOH in SH-SY5Y cells after a 24 h treatment (Figure 3). In a similar experimental approach, shorter treatment (6 and 12 h) of SH-SY5Y cells with coumarins did not prevent the ROS formation supporting the indirect antioxidant effects recorded after 24 h treatment, the time necessary to induce a neuronal antioxidant response (data not shown). Taken together, these results indicate that the coumarins with the highest similarity of chemical structure, such as ESC and DAPH, show both direct and indirect antioxidant effects, suggesting a profile of a bifunctional antioxidant. In particular, the direct antioxidant effects of ESC and DAPH in SH-SY5Y cells suggest their ability to counteract the ROS generated from lipid peroxidation in neurons. These results are consistent with recent studies demonstrating that ESC significantly reduced thiobarbituric acid reactive species formation in rat brain homogenates, a well-established method for measuring lipid peroxidation [15,35]. In addition, other studies employed a treatment time of HepG2 liver cells and V79-4 lung fibroblasts with ESC before the treatment with hydrogen peroxide similar to that adopted in our SH-SY5Y cells (i.e., 24 h) recording both indirect antioxidant and cytoprotective effects and backing our results [36,37].

In order to better evaluate the ability of ESC and DAPH to exert their antioxidant activity at the neuronal level, we measured the TAA, expressed as µmol of trolox equivalent (TE) for mg protein, of cytosolic and membrane-enriched fractions of SH-SY5Y cells treated with 20 μM coumarins for 2 h. At the end of incubation, cytosolic and membrane fractions were separated and submitted to the ABTS assay. Both coumarins significantly enhanced the TAA of SH-SY5Y cell cytosol (untreated vs. treated with ESC: 57.15 ± 7.56 vs. 88.18 ± 13.76 µmol TE/mg protein, *p* < 0.05; untreated vs. treated with DAPH: 88.40 ± 3.84 vs. 135.80 ± 1.19 µmol TE/mg protein, *p* < 0.05), while we did not record antioxidant activity in the membrane fraction (data not shown). These results show the ability of ESC and DAPH to cross the cell membrane and reach the cytoplasm where they exert their antioxidant activity. Remarkably, these results also support the ability of ESC to cross complex biological barriers, such as the intestinal barrier and the BBB in various experimental animal models [21,38].

### 3.2. Effects of ESC and DAPH on Neuronal Antioxidant Response

The coumarins ESC and DAPH showed indirect antioxidant effects in terms of ability to prevent ROS formation, suggesting the activation of neuronal antioxidant responses. We thus investigated whether these antioxidant effects might result from an increase in GSH levels and Nrf2 activation, which play key roles in protecting neuronal cells against oxidative stress. In this context, under quiescent conditions, Nrf2 is complexed in the cytoplasm with the Kelch-like ECH-associated protein 1 (Keap1). In conditions of the breakage of this Nrf2/Keap1 complex, Nrf2 translocates into the nucleus and binds to antioxidant response elements (AREs) in the promoter regions of its target genes, like those for GSH synthesis, activating their transcription [10].

The SH-SY5Y cells were incubated for 24 h with ESC and DAPH [20 µM], and the intracellular GSH levels were then determined using the fluorescent probe MCB. The treatment with ESC, but not DAPH, was shown to significantly increase the GSH levels in SH-SY5Y cells (Figure 4A). In addition, the treatment with ESC (20 µM) at increasing times from 1 to 24 h registered significant decreases in GSH levels after short treatment times (1 and 2 h) and significant increases in GSH levels after long treatment times (12 and 24 h) (Figure 4B).

In parallel, short treatment times of SH-SY5Y cells with ESC and DAPH [20 μM] were also used to evaluate the ability of these coumarins to promote the translocation the Nrf2 into the nucleus and the activation of the Nrf2/ARE binding at this level by Western blotting and ELISA, respectively (Figure 4C,D). Only ESC was shown to significantly increase both the Nrf2 nuclear translocation and Nrf2/ARE binding activity after 1 h of SH-SY5Y cell treatment. Interestingly, given the known involvement of Nrf2 in the transcriptional induction of GSH, we can state that the early activation of Nrf2 elicited by ESC is prodromal of the late increase in GSH. In this regard, the transient decrease of GSH recorded after short treatment times with ESC suggest that it could activate the Nrf2 through its ability to initially induce a partial GSH depletion and changed redox homeostasis. A recent study indirectly demonstrates that *N*-acetyl-l-cysteine, a precursor of cysteine, restores the degradation of GSH elicited by 1 h treatment with ESC in Chinese hamster ovary cells supporting the ability of ESC to initially interfere with the GSH redox cycling and the ratio of GSH to oxidized GSH (GSSG) [39]. Although we did not evaluate the activity of enzymes involved in GSH redox cycling, such as GSH reductase and peroxidase, it is probable that this transient decrease of GSH induces the GR activity which recovers and enhances the GSH levels as well as decreasing the GSSG, as previously reported in primary cortical culture and liver from mice [40,41]. In particular, the ESC induced the GR activity without changing the GPx activity, favoring the GSH level.

This particular profile of the biphasic dose-response (i.e., a hormetic response) appears to be involved in the neuroprotection action of several antioxidants that through their ability to perform Michael additions with thiols present on biological molecules including GSH and the Keap-1 negative repressor of Nrf2 [42]. The hydroxyl groups in ESC position six and seven can improve the ability to perform Michael addition more than DAPH position seven and eight. In this context, other in vitro studies in C2C12 myoblasts and HepG2 liver cells support these findings of ESC’s ability to protect the neuronal cells from oxidative stress through the activation of Nrf2 [36,43]. Remarkably, in a more recent in vivo study, the administration of ESC enhanced the Nrf2 levels and counteracted the mitophagy and mitochondrial apoptosis in hippocampus of mice with cerebral ischemia and reperfusion injury [38].

### 3.3. Neuroprotective Effects of ESC Against Aβ_1–42_ Oligomer-Induced Neuronal Death

Among the studied coumarins, we selected the coumarin ESC that showed both indirect and direct antioxidant effects to evaluate the neuroprotective effects against the toxicity of OAβ_1–42_ using the previous experimental approach of treatments of SH-SY5Y cells with ESC. Several studies suggest that OAβ_1–42_ adheres to neurons and causes oxidative damage of plasma membrane which initiates a cascade of pathological processes that end with neuronal death [44].

We first evaluated the ability of ESC to counteract the neurotoxic events induced by OAβ_1–42_. SH-SY5Y cells were treated with ESC (20 µM) and OAβ_1–42_ (10 µM) for 24 h and the neuronal death was then measured using the fluorescent probe PI. The treatment with ESC was shown to significantly decrease the neuronal death evoked by OAβ_1–42_ (Figure 5A). In parallel, SH-SY5Y cells were treated for 24 h with ESC (20 µM) and then 24 h with OAβ_1–42_ (10 µM). At the end of this combined treatment, ESC significantly prevented the neuronal death induced by OAβ_1–42_ in SH-SY5Y cells (Figure 5B).

Finally, we investigated the ability of ESC to prevent the OS elicited by OAβ_1–42_. SH-SY5Y cells were treated for 24 h with ESC (20 µM) and then treated with OAβ_1–42_ (10 µM) for 3 h. At the end of incubation, the ROS formation was determined using the fluorescent probe DHE. The treatment with ESC was shown to significantly decrease the ROS formation induced by OAβ_1–42_ (Figure 6).

These results demonstrate for the first time the cytoprotective effects of ESC against the neuronal death evoked by OAβ_1–42_. Since recent evidence has demonstrated that ESC lacks chemical properties to inhibit the aggregation of Aβ_25–35_ peptide (the core of the Aβ_1–42_ peptide involved in its aggregation and neurotoxicity), the neuroprotective effects of ESC occurring in concomitant treatment with OAβ_1–42_ could be ascribed to independent neuroprotective mechanisms by its interaction with OAβ_1–42_ [35]. Recent studies indicate that early mitochondrial impairment in the absence of neuronal death by OAβ_1–42_ appears as inhibition of mitochondrial MTT reduction by enhancing MTT formazan exocytosis [45,46]. In this regard, we recorded that the concomitant short treatment of SH-SY5Y cells with ESC for 4 h significantly inhibited the early MTT formazan exocytosis elicited by OAβ_1–42_ (Appendix A). In parallel, we also recorded a recovery of intracellular MTT formazan in the same SH-SY5Y cells indicating the neuroprotective effects of ESC on MTT exocytosis (Appendix A). ESC is therefore likely to counteract early mitochondrial dysfunction mediated by OAβ_1–42_ that triggers the subsequent neuronal death. Intriguingly, it was recently demonstrated that the maintenance of both the mitochondrial metabolism and ATP level by “ATP regulators”, including ESC, via Estrogen Receptor-Related Receptors also showed a significant neuroprotection in the 1-methyl-4-phenyl-1,2,3,6-tetrahydropyridine and rotenone-induced Parkinson’s disease model mice [47].

### 3.4. Effects of ESC on Survival Kinase Pathways

The ability of ESC to prevent both the ROS formation and neuronal death induced by OAβ_1–42_ in SH-SY5Y cells using the same experimental approach of treatment that recorded indirect antioxidant effects and the activation of neuronal antioxidant response suggest that these neuroprotective effects share some upstream signaling pathways. These highlights prompted us to evaluate several kinases, including Erk, Akt and GSK3β, involved in both Nrf2 activation and neuronal survival pathways [48].

The phosphorylation of Erk, Akt and GSK3β at Ser9 kinases was determined after different times of treatment (15, 30, 60 and 120 min) of SH-SY5Y cells with ESC (20 µM) by Western blotting. The treatment of 15 and 30 min with ESC decreased the phosphorylation of Erk kinase while the treatment of 120 min with ESC increased the phosphorylation of Erk kinase (Figure 7A). In parallel, after 120 min of treatment with ESC, we also registered an increase of the Akt kinase phosphorylation (Figure 7B). Further, the same treatment was shown to induce the phosphorylation of GSK3β at Ser9, the inactive form (Figure 7C). In contrast to the nuclear activation of Nrf2 in SH-SY5Y cells after 1 h of treatment with ESC, all the studied kinases showed a significant phosphorylation after 2 h of treatment, suggesting that these kinases are probably not involved in the upstream activation of Nrf2 or neuronal antioxidant response. However, it is plausible that the activation of these kinases contributes to the neuroprotective effects of ESC only in terms of increasing neuronal survival.

To confirm these observations, we used PD98059 (5 µM) and LY294002 [10 µM], inhibitors of Erk and Akt phosphorylation, respectively, during the treatment of SH-SY5Y cells with ESC for 24 h. Cells were then treated for 24 h with OAβ_1–42_ (10 µM). As shown in Figure 8, the neuroprotective effects of ESC against the OAβ_1–42_ induced neuronal death were abrogated by using PD98059 and LY294002. In the same experimental conditions, we also recorded similar data using BSO (400 µM), an inhibitor of GSH synthesis (Figure 8).

These results provide significant proof that Erk and Akt signaling pathways are involved in the neuroprotection mediated by ESC. Recent studies showed the ability of ESC to promote cytoprotective effects in C2C12 myoblasts and human dermal fibroblasts through the phosphorylation of ERK and AKT, respectively, confirming our findings [43,49]. Given the ability of Akt to control glycogen synthesis through phosphorylation and inactivation of GSK3β, we can also speculate that the Akt/GSK3β pathway is involved in the neuroprotective action of ESC.

Activation of GSK3β seems to trigger several pathogenic events in AD, such as hyper-phosphorylation of tau, increased production of Aβ, neuroinflammation and neuronal loss [50].

However, further studies are needed to fully understand the signaling pathways involved in the neuroprotection mediated by ESC. Last, the abolition of the neuroprotective effects of ESC by the inhibition of GSH synthesis also suggests that the induction of GSH at the neuronal level could prove useful as a tool for controlling OS in AD.

## 4. Conclusions

Taken together, these results show the ability of ESC to prevent and counteract OS in SH-SY5Y cells, suggesting its profile of a bifunctional antioxidant. In particular, ESC increases the resistance of the SH-SY5Y cells against OS through the activation of Nrf2 and increase of intracellular GSH. In similar experimental conditions, we further demonstrated that ESC can protect the SH-SY5Y cells from the toxicity evoked by OAβ_1–42_. Although we provide a rationale for the use of ESC as an antioxidant in AD, several limits to effective antioxidant treatment must also be considered. In particular, several studies that used compounds with the ability to directly scavenge the ROS recorded poor antioxidant effects as well as conflicting data in human clinical trials probably because they act on complex oxidative events ongoing with other neurodegenerative events [51,52]. However, more research is necessary to evaluate the therapeutic impact of bifunctional antioxidants, including the coumarins, on neurodegenerative processes in human. Several in vitro and in vivo studies on neuroprotection suggest that the compounds with indirect antioxidant activity through the induction of cytoprotective and antioxidant proteins with long half-lives could ensure a long-term action and resolve neurodegenerative processes already in progress [13].

Among the coumarins, ESC also exhibits a potent inhibitory activity against acetylcholinesterase, butyrylcholinesterase and β-Secretase, advancing its potential profile of multifunctional neuroprotective compound for AD [53]. Our results therefore encourage further research into AD models to explore the efficacy and safety profile of ESC as a novel neuroprotective agent.

## Figures and Tables

**Figure 1 antioxidants-09-00551-f001:**
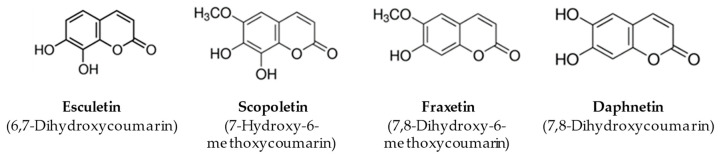
Chemical structure of coumarins.

**Figure 2 antioxidants-09-00551-f002:**
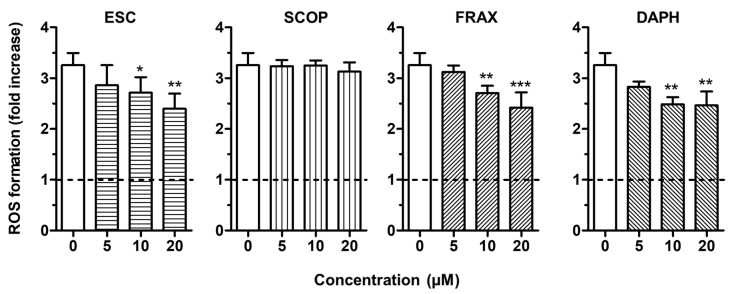
ESC, FRAX and DAPH, but not SCOP, counteract *t*-BuOOH—induced reactive oxygen species (ROS) formation in SH-SY5Y cells. Cells were treated for 2 h with various concentrations of the studied coumarins (5–20 µM) and then with *t*-BuOOH (200 µM) for 30 min. At the end of treatment, ROS formation was determined using the fluorescent probe 2′,7′-dichlorodihydrofluorescein diacetate (DCFH-DA). Data are expressed as fold increases of ROS formation induced by *t*-BuOOH and reported as mean ± SD of four independent experiments (* *p* < 0.05, ** *p* < 0.01 and *** *p* < 0.001 vs. cells treated with *t*-BuOOH at one-way ANOVA with Dunnett post-hoc test).

**Figure 3 antioxidants-09-00551-f003:**
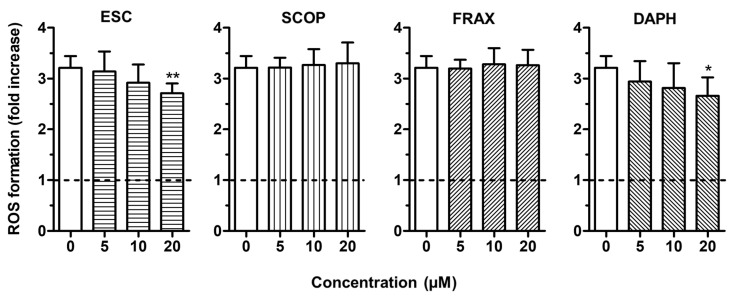
ESC and DAPH, but not SCOP and FRAX, prevent *t*-BuOOH—induced ROS formation in SH-SY5Y cells. Cells were treated for 24 h with various concentrations of the studied coumarins (5–20 µM) and then with *t*-BuOOH (200 µM) for 30 min. At the end of treatment, ROS formation was determined using the fluorescent probe DCFH-DA. Data are expressed as fold increases of ROS formation induced by *t*-BuOOH and reported as mean ± SD of four independent experiments (* *p* < 0.05 and ** *p* < 0.01 vs. cells treated with *t*-BuOOH at one-way ANOVA with Dunnett post-hoc test).

**Figure 4 antioxidants-09-00551-f004:**
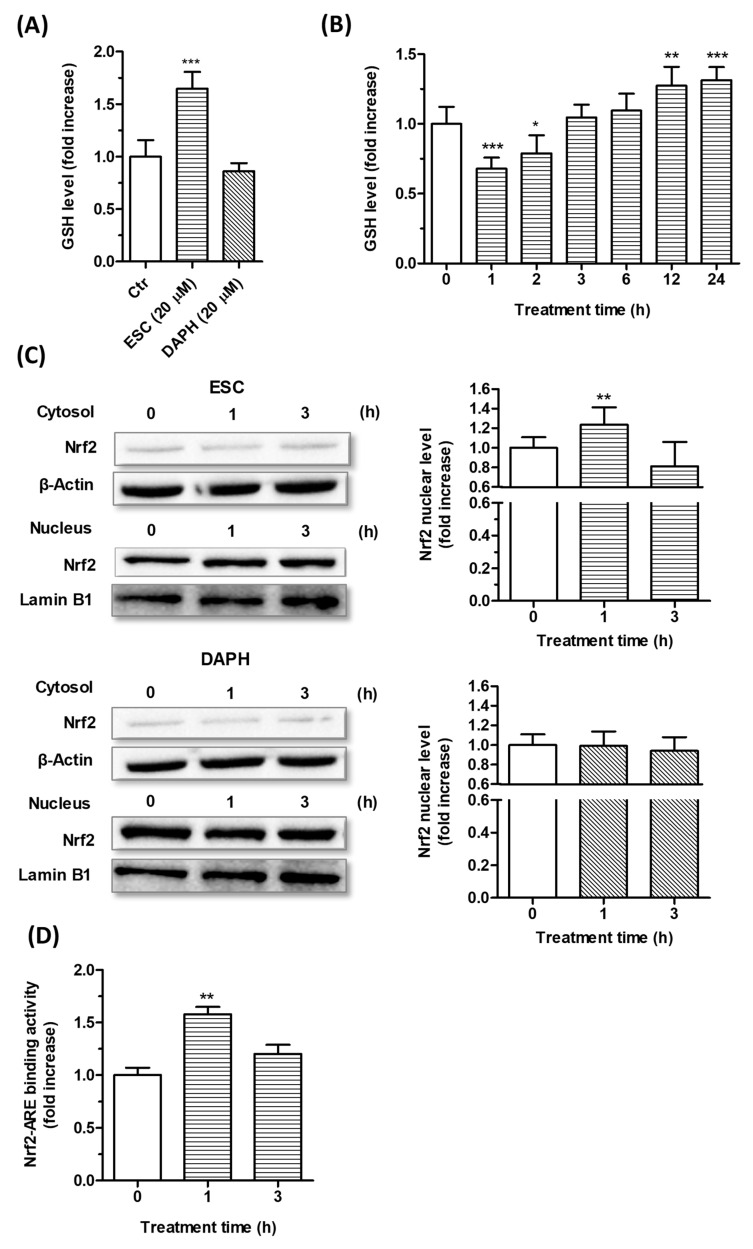
Effects of ESC and DAPH on neuronal antioxidant response in SH-SY5Y cells. (**A**,**B**) Cells were treated with ESC and DAPH (20 µM) for 24 h and ESC (20 µM) for different times. At the end of treatment, the GSH level was measured using the fluorescent probe MCB. (**C**) Cells were incubated with ESC and DAPH (20 µM) for 1 and 3 h. At the end of incubation, the Nrf2 nuclear level was analyzed by Western blotting. (**D**) Cells were treated with ESC (20 µM) for 1 and 3 h. At the end of treatment, Nrf2-ARE binding activity was determined by ELISA assay. Data are expressed as fold increases and reported as mean ± SD of three independent experiments (* *p* < 0.05, ** *p* < 0.01 and *** *p* < 0.001 vs. untreated cells at one-way ANOVA with Dunnett post-hoc test).

**Figure 5 antioxidants-09-00551-f005:**
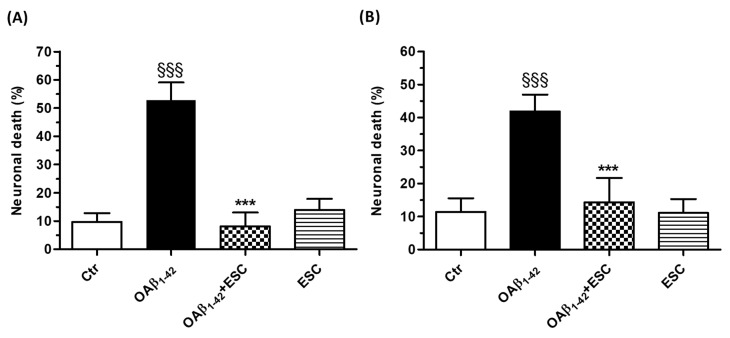
ESC counteracts and prevents the neuronal death induced by OAβ_1–42_ in SH-SY5Y cells. (**A**) Cells were treated with ESC (20 µM) and OAβ_1–42_ (10 µM) for 24 h. (**B**) Cells were treated with ESC (20 µM) for 24 h and then with OAβ_1–42_ (10 µM) for 24 h. At the end of treatment, the neuronal death was determined using the fluorescent probe PI. Data are expressed as percentages of dead cells and reported as mean ± SD of three independent experiments (^§§§^
*p* < 0.001 vs. untreated cells and *** *p* < 0.001 vs. cells treated with OAβ_1–42_ at one-way ANOVA with Bonferroni post-hoc test). Scale bars: 100 µM.

**Figure 6 antioxidants-09-00551-f006:**
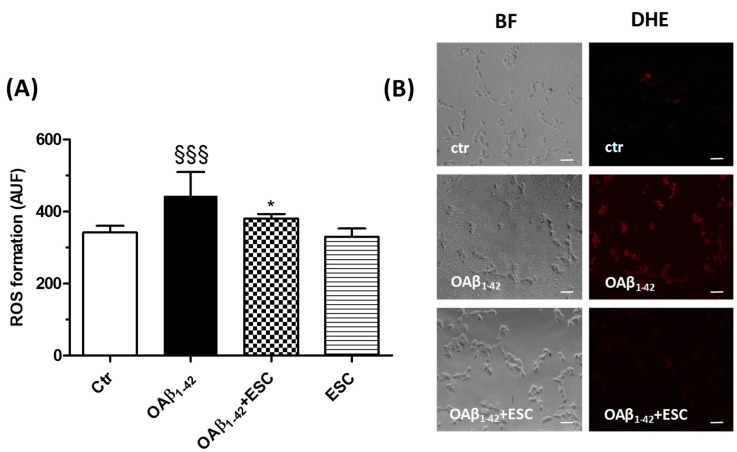
ESC prevents the ROS formation induced by OAβ_1–42_ in SH-SY5Y cells. (**A**) Cells were treated with ESC (20 µM) for 24 h and then with OAβ_1–42_ (10 µM) for 3 h. At the end of treatment, the ROS formation was determined using the fluorescent probe DHE. (**B**) Representative bright field (BF) and DHE fluorescence images. Data are expressed as AUF and reported as mean ± SD of three independent experiments (^§§§^
*p* < 0.001 vs. untreated cells, * *p* < 0.05 vs. cells treated with OAβ_1–42_ at one-way ANOVA with Bonferroni post-hoc test). Scale bars: 100 µM.

**Figure 7 antioxidants-09-00551-f007:**
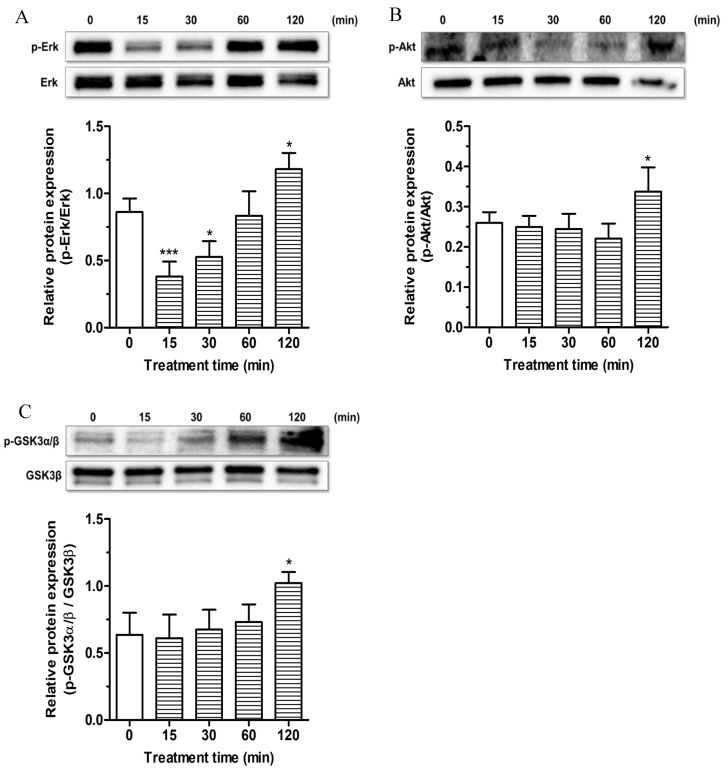
Effects of ESC on survival kinase pathways in SH-SY5Y cells. Cells were treated with ESC (20 µM) for different times. At the end of treatment, the phosphorylation of Erk, Akt and GSK3α/β kinase levels was measured by Western blotting. Data are expressed as the ratio between phosphorylated form and total protein expression and reported as mean ± SD of three independent experiments (* *p* < 0.05 and *** *p* < 0.001 vs. untreated cells at one-way ANOVA with Dunnett post-hoc test).

**Figure 8 antioxidants-09-00551-f008:**
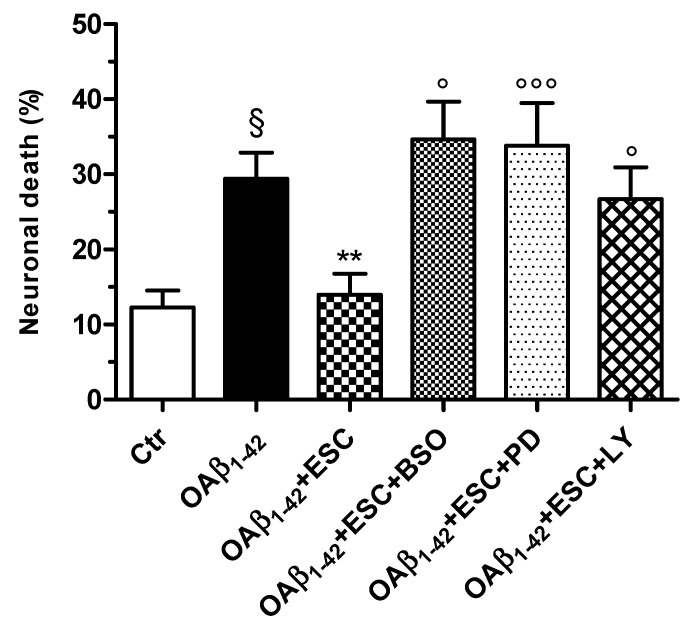
Neuroprotective effects of ESC are abrogated by the specific inhibitors BSO, PD98059 and LY294002 in SH-SY5Y cells. Cells were treated with ESC (20 µM) and the specific inhibitors BSO (400 µM), PD98059 (5 µM) and LY294002 (10 µM) for 24 h. Subsequently, cells were treated with OAβ_1–42_ (10 µM) for 24 h. At the end of treatment, the neuronal death was determined using the fluorescent probe PI. Data are expressed as percentage of dead cells and reported as mean ± SD of four independent experiments (^§^
*p* < 0.05 vs. untreated cells; ** *p* < 0.01 vs. cells treated with OAβ_1–42_; ° *p* < 0.05 and °°° *p* < 0.001 vs. cells treated with OAβ_1–42_ + ESC at one-way ANOVA with Bonferroni post-hoc test).

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
