# Peer review of "Esculetin as a Bifunctional Antioxidant Prevents and Counteracts the Oxidative Stress and Neuronal Death Induced by Amyloid Protein in SH-SY5Y Cells"

_antioxidants, 2020, doi:10.3390/antiox9060551_

Round 1
Reviewer 1 Report
The authors’ work reports the efficacy of several coumarin family compounds, and based on their findings the most efficacious in ameliorating oxidative status is esculetin. The authors discuss esculetin ability to directly scavenge reactive oxygen species (ROS) and to trigger Nrf2 pathway, enhancing cells endogenous antioxidant response. In parallel the authors report the efficacy of esculetin in arresting cell death and ROS formation upon challenging with oligomeric amyloid-beta peptide.
Overall, the methods are adequate to the questions posed by the authors and the subject is timely. The manuscript needs some attention for a few typos. The introduction is by times vague and specific topics will follow for the authors to address. In the results and discussion section of the manuscript the reviewer considers several major issues need to be addressed.
The reviewer encourages the authors to reconsider their conclusions addressing the comment as follows.
In detail:
1 - P1; L38: Considering hyperphosphorylated tau protein as a hallmark for AD is a common misconception in the literature. Moreover, the role of hyperphosphorylation in tau aggregation is debatable. It is more correct to state that neurofibrillary tangles composed of paired helical filaments of aggregated tau is in fact one hallmark of AD.
2 - P2; L45: Which enzyme? There are several enzymes involved in glutathione cycling.
3 - P2; L51: Of which neurodegenerative events and which type of models (in vitro, animals?)
4 - P2: Indicate in the legend of figure one the names of the chemical entities in full as well the IUPAC names.
5 - Figures: Indicate for each figure the exact number of independent experiments in the figure legends, rather than always stating “at least 3 independent experiments”.
6 - P6-7: The difference between figures 2 and 3 is that: (a) in the former the authors claim that some coumarins counteract ROS formation, and (b) in the latter some coumarins prevent ROS formation. Experimentally the reviewer understands such difference in the conclusions is allowed by changing the order of incubation between the antioxidants and the oxidative agent. From the figure legends the authors seem to base their conclusions on short-term vs long-term incubation times with the coumarins. To the extent of incubation time, the reviewer considers that the authors can only infer about the type of action, meaning, being direct ROS scavenging for figure 2 and activation of secondary antioxidant pathways for figure 3. The claim made in heading 3.1. is supported by the results, however in the text when describing and distinguishing counteracting vs preventive effects, the reviewer considers there is not enough data to support the claim.
7 - P8; L318-321: This description of results may need to be revisited after an adequate normalization marker is shown for the nuclear fractions, as noted in the following comment.
8 - P9; Figure 4: Normalization of the nuclear Nrf2 should not be done by a cytosolic marker. The reviewer suggests normalizing with a nuclear marker such as TATA-box or other. In fact, the labelling of actin in the nuclear fractions just indicates that the nuclear fractions are not pure. In regard to purity, once the authors normalize with a nuclear marker, the same marker should also be used to label the cytosolic fractions to access the purity of the cytosolic fractions.
9 - P8; L324-326: Can the authors discuss how ESC can putatively decrease GSH partially? Is it a matter of direct interaction or secondary response to the presence of an exogenous antioxidant through enzymatic glutathione cycling? It seems unclear how the presence of ESC can trigger Nrf2 translocation based in GSH transient depletion and redox imbalance, since ESC immediate action is direct ROS scavenging and therefore there would be no interference with Keap1-Nrf2 complex. Thus, it seems unlikely the hypothetical mechanism happens like the authors formulate. The reviewer suggest that the authors demonstrate in the same time-range what are the effects of ESC alone in the levels of ROS and eventually the activities of glutathione peroxidase and glutathione reductase.
10 - P10; L361-362: There are no representative images for this figure (5C) in the manuscript. And it is not cited in the text as well.
11 - P13; L450-451: Oxidative stress is a common event in several neurodegenerative diseases. On the other hand, the use of antioxidants as therapies in neurodegeneration revealed over time to have poor translation potential to the clinic. Could the authors elaborate on these issues in their conclusions?
Author Response
Response to Reviewer 1 Comments
We thank the reviewer for the critical appraisal and suggestions and we modified the text in accordance with scientific statements.
1 - P1; L38: Considering hyperphosphorylated tau protein as a hallmark for AD is a common misconception in the literature. Moreover, the role of hyperphosphorylation in tau aggregation
is debatable. It is more correct to state that neurofibrillary tangles composed of paired helical filaments of aggregated tau is in fact one hallmark of AD.
ANSWER. We changed “hyperphosphorylation tau protein” to “neurofibrillary tangles” (Introduction section 1).
2 - P2; L45: Which enzyme? There are several enzymes involved in glutathione cycling.
ANSWER. We improved this sentence. Since several studies reported an impairment of GSH synthetase (Mandal et al. 2015, reference 7) we changed “enzyme activity” to “GSH synthetase” and “GSH redox cycle” to “GSH synthesis” (Introduction section 1).
3 - P2; L51: Of which neurodegenerative events and which type of models (in vitro, animals?)
ANSWER. We added new information on the inflammation events evaluated in these experimental models. In particular, we wrote that it has been demonstrated that Nrf2 participates in inflammation, a critical process in the pathogenesis of neurodegenerative diseases, in both in in vitro and in vivo neuroinflammation models. In particular, Nrf2 represses the inflammatory responses mediated by nitric oxide synthase, interleukin-6, TNF-α (Introduction section 1).
4 - P2: Indicate in the legend of figure one the names of the chemical entities in full as well the IUPAC names.
ANSWER. We added the full and IUPAC names in figure 1.
5 - Figures: Indicate for each figure the exact number of independent experiments in the figure legends, rather than always stating “at least 3 independent experiments”.
ANSWER. We added the exact number of experiments in all figure legends.
6 - P6-7: The difference between figures 2 and 3 is that: (a) in the former the authors claim that some coumarins counteract ROS formation, and (b) in the latter some coumarins prevent ROS formation. Experimentally the reviewer understands such difference in the conclusions is allowed by changing the order of incubation between the antioxidants and the oxidative agent. From the figure legends the authors seem to base their conclusions on short-term vs long-term incubation times with the coumarins. To the extent of incubation time, the reviewer considers that the authors can only infer about the type of action, meaning, being direct ROS scavenging for figure 2 and activation of secondary antioxidant pathways for figure 3. The claim made in heading 3.1. is supported by the results, however in the text when describing and distinguishing counteracting vs preventive effects, the reviewer considers there is not enough data to support the claim.
ANSWER. We thank the Reviewer for the comments on our experimental approach to evaluate the direct and indirect antioxidant activity. As reported in section 3.1, we evaluated the direct antioxidant activity of coumarins after 2 h treatment of SH-SY5Y cells. In this experimental condition it is plausible to measure the direct scavenger activity of the coumarins present in the cells against free radicals. We also treated the SH-SY5Y cells with coumarins for 24 h to evaluate the indirect antioxidant activity. The long term treatment of 24 h is necessary to give the coumarins time to induce a neuronal antioxidant response and exert an indirect antioxidant activity. In this regard we also evaluated the antioxidant activity of coumarins after shorter treatment times than the 24 h treatment, such as 6 and 12 h of treatment. The treatment of SH-SY5Ycells with coumarins for 6 or 12 h did not prevent the ROS formation induced by t-BuOOH, supporting the indirect antioxidant activity recorded with coumarins after 24 h treatment, the time necessary to increase the intracellular antioxidant defense. We added the results obtained with shorter treatments (6 and 12 h) of SH-SY5Y cells with coumarins to support the claim of indirect antioxidant activity in section 3.1.
7 - P8; L318-321: This description of results may need to be revisited after an adequate normalization marker is shown for the nuclear fractions, as noted in the following comment.
ANSWER. We replied to this comment together with 8-P9.
8 - P9; Figure 4: Normalization of the nuclear Nrf2 should not be done by a cytosolic marker. The reviewer suggests normalizing with a nuclear marker such as TATA-box or other. In fact, the labelling of actin in the nuclear fractions just indicates that the nuclear fractions are not pure. In regard to purity, once the authors normalize with a nuclear marker, the same marker should also be used to label the cytosolic fractions to access the purity of the cytosolic fractions.
ANSWER. We thank the Reviewer for the suggestions reported in both 7-P8 and 8-P9. Certainly, actin is usually used as a cytoplasmic marker. Anyway, several Authors recognize that cytoskeletal proteins populate the cell nucleus and have fundamental functions also in this cellular compartment. About 20% of cellular actin is present in the nucleus and a considerable fraction undergoes dynamic nucleocytoplasmic transport in and out of the nucleus (Percipalle and Vartiainen. Cytoskeletal proteins in the cell nucleus: a special nuclear actin perspective. Mol Biol Cell. 2019, 30, 1781-1785). Inside the nucleus, actin forms a reticulated pattern, implying that actin filament is required for nuclear formation (Krauss et al. Nuclear actin and protein 4.1: essential interactions during nuclear assembly in vitro. Proc. Natl Acad. Sci. USA, 2003, 100, 10752-10757).
Maintenance of the nuclear actin levels by active nuclear transport are directly connected to nuclear function, and sufficient nuclear actin levels are required (Falahzadeh et al. The potential roles of actin in the nucleus. Cell J. 2015, 17, 7-14; Stuven et al. Exportin 6: a novel nuclear export receptor that is specific for profilin.actin complexes. EMBO J. 2003, 22, 5928–5940; Dopie et al. Active maintenance of nuclear actin by importin 9 supports transcription. Proc Natl Acad Sci USA, 2012, 109, E544–E552). Beside the structural role of actin in the nuclear matrix (Clubb and Locke. Peripheral nuclear matrix actin forms perinuclear shells. J Cell Biochem. 1998, 70, 240-251), this protein is involved in several nuclear processes; it is part of the chromatin remodeling complex (Shen et al. A chromatin remodeling complex involved in transcription and DNA processing. Nature. 2000, 406, 541-544); it is associated with the transcription processes (Obrdlik et al. The function of actin in gene transcription. Histol Histopathol. 2007, 22, 1051-1055; Zheng et al. Nuclear actin and actin-binding proteins in the regulation of transcription and gene expression. FEBS J. 2009, 276, 2669-2685); and with ribonucleoproteins, as demonstrated by mass spectrometry and immunoreactivity experiments that have shown that β-actin is the nuclear isoform of actin associated with heterogeneous nuclear ribonucleoproteins (hnRNPs) and RNA polymerase complexes (Hofmann et al. Actin is part of pre-initiation complexes and is necessary for transcription by RNA polymerase II. Nat Cell Biol. 2004, 6, 1094-1101; Jonsson et al. Recovery of gel-separated proteins for in-solution digestion and mass spectrometry. Anal Chem. 2001, 73, 5370-5377; Hu et al. A role for beta-actin in RNA polymerase III transcription. Genes Dev. 2004,18, 3010-3015). In view of these considerations, we can affirm that normalizing nuclear protein with β-actin can be considered as an option, and the labelling of our nuclear samples is not due to a low purity grade.
9 - P8; L324-326: Can the authors discuss how ESC can putatively decrease GSH partially? Is it a matter of direct interaction or secondary response to the presence of an exogenous antioxidant through enzymatic glutathione cycling? It seems unclear how the presence of ESC can trigger Nrf2
translocation based in GSH transient depletion and redox imbalance, since ESC immediate action is direct ROS scavenging and therefore there would be no interference with Keap1-Nrf2 complex. Thus, it seems unlikely the hypothetical mechanism happens like the authors formulate. The reviewer suggest that the authors demonstrate in the same time-range what are the effects of ESC alone in the levels of ROS and eventually the activities of glutathione peroxidase and glutathione reductase.
ANSWER. We thank the Reviewer for these comments. In this regard, a recent study indirectly demonstrates that N-acetyl-L-cysteine, a precursor of cysteine, restores the degradation of GSH elicited by 1 h treatment with esculetin in Chinese hamster ovary cells supporting the ability of esculetin to initially interfere with the GSH redox cycling and the ratio of GSH to oxidized GSH (GSSG) (Shinde et al. Fluorescence "off" and "on" signalling of esculetin in the presence of copper and thiol: a possible implication in cellular thiol sensing. Photochem Photobiol Sci. 2018, 17, 1197‐1205). Although we did not evaluate the activity of enzymes involved in GSH redox cycling, such as GSH reductase and peroxidase, it is probable that this transient decrease of GSH induces the GR activity which recovers and enhances the GSH levels as well as decreasing the GSSG, as previously reported in primary cortical culture and liver from mice (Lee et al. Esculetin inhibits N-methyl-D-aspartate neurotoxicity via glutathione preservation in primary cortical cultures. Lab Anim Res. 2011, 27, 259‐263; Martin-Aragón et al. Effects of the antioxidant (6,7-dihydroxycoumarin) esculetin on the glutathione system and lipid peroxidation in mice. Gerontology. 1998, 44, 21‐25). In particular, the ESC induced the GR activity without changing the GPx activity favoring the GSH level. All this information has been added as a discussion of the data obtained in section 3.2.
10 - P10; L361-362: There are no representative images for this figure (5C) in the manuscript. And it is not cited in the text as well.
ANSWER. We apologize for this mistake. We therefore deleted this information from the legend of figure 5.
11 - P13; L450-451: Oxidative stress is a common event in several neurodegenerative diseases. On the other hand, the use of antioxidants as therapies in neurodegeneration revealed over time to have poor translation potential to the clinic. Could the authors elaborate on these issues in their
conclusions?
ANSWER. We thank the Reviewer for stimulating the discussion of the antioxidant therapy in AD. We therefore emphasized this aspect, discussing the potential disadvantages and advantages of bifunctional antioxidants in our context in the conclusion section 4. In particular, we reported that although we provide a rationale for the use of esculetin as an antioxidant in AD, several limits to effective antioxidant treatment must also be considered. In particular, several studies that used compounds with the ability to directly scavenge the ROS recorded poor antioxidant effects as well as conflicting data in human clinical trials probably because they act on complex oxidative events ongoing with other neurodegenerative events (Pitchumoni and Doraiswamy. Current status of antioxidant therapy for Alzheimer's Disease. J Am Geriatr Soc. 1998, 46, 1566‐1572; LLoret et al. Is antioxidant therapy effective to treat alzheimer's disease? Free Rad Antiox. 2011, 1, 8-14.). However, more research is necessary to evaluate the therapeutic impact of bifunctional antioxidants, including the coumarins, on neurodegenerative processes in human. Several in vitro and in vivo studies on neuroprotection suggest that the compounds with indirect antioxidant activity through the induction of cytoprotective and antioxidant proteins with long half-lives could ensure a long-term action and resolve neurodegenerative processes already in progress (Dinkova-Kostova and Talalay. Direct and indirect antioxidant properties of inducers of cytoprotective proteins. Mol Nutr Food Res. 2008, 52, S128-138).
Reviewer 2 Report
The work submitted by Prucolli and colleagues deals with the molecular mechanisms that might make esculetin a valuable drug for treatment of Alzheimer’s disease. They describe the antioxidative property of the plant-derived compound by administering it to SH-SY5Y cells, a widely used secondary cell line in the field. The presented experiments are in general soundly planned and conducted and the usage of different concentrations is appreciated as well as the initial co-testing of other coumarins.
I mainly have some minor points that should be addressed before the manuscript might be eligible for publication:
- Some minor language mistakes have to be corrected such as “oxidants” instead of “oxidant” (line 59) or plant species names which should be given in italics.
- After the abbreviation is given the first time, the authors should stick to it: e.g. “esculetin” in lines 66, 69, and 71 (abbrev. is already given in line 64).
- Some important references seem to be missing as for example Ali, Jannat…AsianPacJ Trop Med 2016 and Shinde et al., Photochem Photobiol Sci 2018 or Martin-Aragon…, Gerontology 1998. These already report on some important aspects in regard to esculetin such as interference with GSH in vivo or Ache/BChe inhibitory function.
- Some details of the material/methods section are missing: e.g. if the medium used here contained phenolred or not (which is important for SH-SY5Y cells), solvent of the coumarins, explanation for Trolox, solvent of DCFH-DA and volume BuOOH.
- As the total cell number added per well seems rather low for SH-SY5Y cells, I would love to have exemplary microscopic pictures of the main findings (besides the fluorescence picture in Fig. 6). This will allow the reader to estimate the general condition of the cells within the experiments. Moreover, the authors report on that they used Trolox as a sort of gold standard for the intrinsic antioxidant activity. I guess this is a good idea and would suggest to also add this as a comparison control to all other experimental settings. In addition, the concentration of Trolox should be given.
- As the authors investigated phosphorylated kinases, it would have been mandatory to add phosphatase inhibitors to the cell lysates. Has this been done?
- Stripping procedure of the western blots should be described. This is of importance as p-ERK and ERK, for example, do not differ so much in molecular weight. If the stripping procedure is not perfectly conducted and the same secondary antibody used, then carry-over of the first signal has to be expected.
- Which filters were used for method described in 2.12?
- How was PI measured (2.13)?
- I guess the data for the intrinsic anti-oxidant effect are missing?! In the first paragraph of 3.1 only data from the ABTS experiment are mentioned but not from DPPH assay? I would suggest also to add these data as a figure or at least in a supplementary figure.
- The authors describe that they used 20 µM of the coumarins on cells as this concentration did not affect viability. Was this the highest non-toxic concentration or was it juts the highest tested concentration?
- N-numbers should be given for each experiment.
- The amount of A-beta peptide to evoke cellular toxicity is rather high. Can the authors give a rationale for this?
- 10% of toxicity just evoked by the solvent (Figure 5A/B) seems rather high – what kind of solvent was used and in which vol/vol %?
- Quality of the blot in Figure 7 (p-Akt/Akt) is not really high. Especially, the p-form would be hard to quantify in the shown quality.
- The authors could be a little bit more critical about “just” using this cell model. It is well accepted in the field; however, it is still just a secondary cancer cell line. Using primary neurons would have added way more meaning to the data.
Author Response
Response to Reviewer 2 Comments
We thank the reviewer for the critical appraisal and suggestions and we modified the text in accordance with scientific statements.
- Some minor language mistakes have to be corrected such as “oxidants” instead of “oxidant” (line 59) or plant species names which should be given in italics.
ANSWER. We corrected minor mistakes.
- After the abbreviation is given the first time, the authors should stick to it: e.g. “esculetin” in lines 66, 69, and 71 (abbrev. is already given in line 64).
ANSWER. We added the abbreviations for esculetin.
- Some important references seem to be missing as for example Ali, Jannat…AsianPacJ Trop Med 2016 and Shinde et al., Photochem Photobiol Sci 2018 or MartinAragon…, Gerontology 1998. These already report on some important aspects in regard to esculetin such as interference with GSH in vivo or Ache/BChe inhibitory function.
ANSWER. We thank the Reviewer for the suggestions. The reference Ali et al. Asian Pac J Trop Med. 2016 was already present in conclusion section 4 (reference 54). The references Shinde et al. Photochem Photobiol Sci. 2018 and Martin-Aragón et al. Gerontology. 1998 have been added in result section 3.2 (references 39 and 41).
- Some details of the material/methods section are missing: e.g. if the medium used here contained phenolred or not (which is important for SH-SY5Y cells), solvent of the coumarins, explanation for Trolox, solvent of DCFH-DA and volume BuOOH.
ANSWER. We added the missing details for medium, coumarins, DCFH-DA and t-BuOOH. In particular, we integrated with more informations the details for coumarins. Since coumarins are soluble in organic solvent, we prepared stock solutions of coumarins in DMSO then we diluted them in complete medium to obtain the desired concentrations of coumarins in a maximum of 0.1% DMSO (method section 2.3).
We gave a wrong description for the use of trolox in method section 2.4 (determination of intrinsic antioxidant activity). We therefore corrected this mistake because the antioxidant activity was expressed as concentration of coumarin able to decolorize 50% of the ABTS radical. We used the trolox only to do a standard curve to extrapolate the micromole of Trolox equivalent antioxidant activity per milligram protein (µmolTE/mg protein) of membrane and cytosolic fractions. We added this information in method section 2.6 (determination of antioxidant coumarins in membrane and cytosolic fractions)
- As the total cell number added per well seems rather low for SH-SY5Y cells, I would love to have exemplary microscopic pictures of the main findings (besides the fluorescence picture in Fig. 6). This will allow the reader to estimate the general condition of the cells within the experiments. Moreover, the authors report on that they used Trolox as a sort of gold standard for the intrinsic antioxidant activity. I guess this is a good idea and would suggest to also add this as a comparison control to all other experimental settings. In addition, the concentration of Trolox should be given.
ANSWER. We used a low number of SH-SY5Y cells to prevent their growth in cluster that give artifacts during the analysis with the fluorescence microscopy. We reported in Figure 6B the representative images of different experimental conditions, such as control (untreated cells), OAβ1-42 (cells treated with OAβ1-42), OAβ1-42 and ESC (cells treated with OAβ1-42) and ESC (cells treated with ESC). The other pictures used for analysis were very similar to images selected in Figure 6B. We think that other images do not improve our findings.
As reported in reply to comment 4, we apologize for the mistake regarding the trolox in determination of intrinsic antioxidant activity. We therefore corrected this mistake because the antioxidant activity was expressed as concentration of coumarin able to decolorize 50% of the ABTS radical (method section 2.4). We used the trolox only in determination of antioxidant activity of coumarins in membrane and cytosolic fractions (method section 2.6). In this regard, we added new informations such as the range of trolox concentrations (from 50 to 150 µM) used to do the standard curve.
We did not use the trolox, a water-soluble analog of vitamin E, as positive control in our experiments because it is not a bifunctional antioxidant. The trolox retains only the scavenger activity of vitamin E.
- As the authors investigated phosphorylated kinases, it would have been mandatory to add phosphatase inhibitors to the cell lysates. Has this been done?
ANSWER. We always use a cocktail of protease/phosphatase inhibitors [100X] to prevent protein degradation and dephosphorylation by proteases and phosphatases present in the cell lysates. We added this information in method section 2.9.
- Stripping procedure of the western blots should be described. This is of importance as p-ERK and ERK, for example, do not differ so much in molecular weight. If the stripping procedure is not perfectly conducted and the same secondary antibody used, then carry-over of the first signal has to be expected.
ANSWER. We apologize for this lack. In particular, we use a consolidated stripping procedure. This procedure was repeated only once for each membrane, limiting the interference with the first signal. In particular, the membrane was incubated with stripping buffer on plate shaker for 1 hour at room temperature. The stripping buffer contains β-mercaptoethanol (18.3 μM), SDS (69.35 mM), and Tris·HCl (62.5 mM), at pH 6.7. After the incubation, the membrane was washed with TBST for at least 30 min in order to eliminate all the stripping solution and then incubated in block solution. We added the informations of stripping buffer in method section 2.9.
- Which filters were used for method described in 2.12?
ANSWER. We used TRITC filters (EX 535/50, BS 575, EM 590LP). We added this information in method section 2.12.
- How was PI measured (2.13)?
ANSWER. We measured the neuronal death using the fluorescence microscope Eclipse Ti-E (Nikon Instruments Spa). In particular, total cells were counted in bright field then only the red dead cells were counted using TRITC filters (EX 535/50, BS 575, EM 590LP). Last, data were expressed as percentages of dead cells versus total cells. We added these informations in method section 2.13.
- I guess the data for the intrinsic anti-oxidant effect are missing?! In the first paragraph of 3.1 only data from the ABTS experiment are mentioned but not from DPPH assay? I would suggest also to add these data as a figure or at least in a supplementary figure.
ANSWER. We added a supplementary figure with the results of DPPH assay (Figure 1S). These results were similar to those obtained with ABTS assay.
- The authors describe that they used 20 µM of the coumarins on cells as this concentration did not affect viability. Was this the highest non-toxic concentration or was it juts the highest tested concentration?
ANSWER. The concentration 20 µM was the highest concentration not associated to neurotoxicity in SH-SY5Y cells.
- N-numbers should be given for each experiment.
ANSWER. We added the exact number of experiments in all figure legends.
- The amount of A-beta peptide to evoke cellular toxicity is rather high. Can the authors give a rationale for this?
ANSWER. We thank the reviewer for raising this aspect. Concentrations of Aβ1-42 peptide oligomers lower than 10 µM (not less than 5 µM) can exert neuronal death in SH-SY5Y cells after 24 h of treatment. However, to evaluate early neurotoxicity events after short treatment times (2-4 h), such as ROS formation and mitochondrial impairment, is necessary to use 10 µM of Aβ1-42 peptide oligomers to get an appreciable signal of fluorescence or absorbance. We therefore uniformed all experiments to 10 µM of Aβ1-42 peptide oligomers.
- 10% of toxicity just evoked by the solvent (Figure 5A/B) seems rather high – what kind of solvent was used and in which vol/vol %?
ANSWER. We would like to thank the Reviewer for this question. In control cells we added only DMSO at 0.1% (vol/vol), the same solvent used to prepare the stock solutions of coumarins. However, the 10% level falls within the experimental variability considering the microscope procedure.
- Quality of the blot in Figure 7 (p-Akt/Akt) is not really high. Especially, the p-form would be hard to quantify in the shown quality.
ANSWER. We agree with the Reviewer that the quality of the blot for p-Akt/Akt is not really high. However, the Quantify One Analysis Software (Bio-Rad Laboratories S.r.L., Hercules, CA, USA) allowed to select the amplitude of the signal peak during the analysis of these images.
- The authors could be a little bit more critical about “just” using this cell model. It is well accepted in the field; however, it is still just a secondary cancer cell line. Using primary neurons would have added way more meaning to the data.
ANSWER. We thank the Reviewer for this comment. Certainly, rodent primary neurons could add more meaning to the data. However, we initially chose the SH-SY5Y cell line because we needed a lot of cells to simultaneously evaluate several coumarins. After the first selection of coumarins with the best antioxidant activity we preferred to continue with SH-SY5Y cells to maintain the same experimental conditions such as coumarin concentrations and treatment times. Since SH-SY5Y cells are human-derived, they may have some advantages for assessing the ability of compounds to induce antioxidant endogenous proteins, such as the expression of human specific proteins and protein isoforms that would not be inherently present in rodent primary cultures (Kovalevich and Langford. Considerations for the use of SH-SY5Y neuroblastoma cells in neurobiology. Methods Mol Biol. 2013, 1078, 9‐21).
Round 2
Reviewer 1 Report
The authors effort in replying to the reviewers' comments and suggestions is acknowledged and is considered satisfactory for most of the raised topics. Even though the authors provide a reasonable amount of literature to support their rationale for the use of actin as loading control for nuclear fractions in immunoblots, the reviewer considers that the authors need to address this issue experimentally. The authors argue that 20% of the cellular actin is nuclear, the is nucleocytoplasmic transport of the protein and their are structural and functional correlates between actin and nuclear processes. However, other than merely raising fraction purity issues, the question of whether actin truly represents a nuclear normalization marker remains a caveat that needs to be experimentally addressed, even more taking in consideration that in Fig. 4C the levels of actin in the nuclear fraction significantly vary between incubation times. Would this observed difference mean that less nuclear extract was used in the SDS-PAGE or on the other hand that ESC affects nucleocytoplasmic transport of actin, therefore making it an unreliable loading control? The reviewer encourages the authors to experimentally address this issue before resubmission.
Author Response
Response to Reviewer 1 Comments
The authors effort in replying to the reviewers' comments and suggestions is acknowledged and is considered satisfactory for most of the raised topics. Even though the authors provide a reasonable amount of literature to support their rationale for the use of actin as loading control for nuclear fractions in immunoblots, the reviewer considers that the authors need to address this issue experimentally. The authors argue that 20% of the cellular actin is nuclear, the is nucleo cytoplasmic transport of the protein and their are structural and functional correlates between actin and nuclear processes. However, other than merely raising fraction purity issues, the question of whether actin truly represents a nuclear normalization marker remains a caveat that needs to be experimentally addressed, even more taking in consideration that in Fig. 4C the levels of actin in the nuclear fraction significantly vary between incubation times. Would this observed difference mean that less nuclear extract was used in the SDS-PAGE or on the other hand that ESC affects nucleocytoplasmic transport of actin, therefore making it an unreliable loading control? The reviewer encourages the authors to experimentally address this issue before resubmission.
We thank the reviewer for the critical appraisal and we modified the results in Figure 4C in accordance with scientific statements. In particular, we stripped all the membranes with nuclear proteins and reprobed with Lamin B1 (D9V6H) Rabbit mAb (Cell Signaling, Danvers, USA), a nuclear protein marker. The new data were then analyzed by densitometry. All these informations were added in Figure 4C.
Reviewer 2 Report
I appreciate that the authors thoroughly worked on my comments and suggestions. They could nearly answer all my open questions sufficiently. Nevertheless, I would like to insist on showing at least one exemplary light microscopic picture of SH-SY5Y cells treated with ESC and with solvent control. Figure 6 only shows a fluorescent signal and does not allow to evaluate general condition of the cell culture. This would clearly improve the findings by e.g. reporting on unaltered cell morphology or lenghtened neurites etc. I would also ask to give the n number of the experiments. If n number and number of experiments is the same, then adding once the term "n=4" in the figure legends would be sufficient.
Author Response
Response to Reviewer 2 Comments
I appreciate that the authors thoroughly worked on my comments and suggestions. They could nearly answer all my open questions sufficiently. Nevertheless, I would like to insist on showing at least one exemplary light microscopic picture ofSH-SY5Y cells treated with ESC and with solvent control. Figure 6 only shows a fluorescent signal and does not allow to evaluate general condition of the cell culture. This would clearly improve the findings by e.g. reporting on unaltered cell morphology or lenghtened neurites etc. I would also ask to give the n number of the experiments. If n number and number of experiments is the same, then adding once the term "n=4" in the figure legends would be sufficient.
We thank the reviewer for the suggestions and we modified the Figure 6B in accordance with scientific statements. We added new representative images of both bright field and DHE fluorescence in Figure 6B. Further, we improved the methods in section 2.12. In particular, we added informations about the microscope Eclipse Ti-E and equipment as well as the criteria for measuring fluorescence intensity. It is measured from an area corresponding to 20 cells in at least five different random areas for each experiment. We performed three independent experiments. All these informations were added in Figure 6B and section 2.12.
Round 3
Reviewer 1 Report
The reviewer appreciates that the authors significantly improved the quality of the manuscript addressing all previously raised issues.